# Inexpensive Millimeter-Wave Communication Channel Using Glow Discharge Detector and Satellite Dish Antenna

**Lidor Kahana [1,*][iD], Daniel Rozban [1], Moshe Gihasi [1], Amir Abramovich [1,*][iD], Yitzhak Yitzhaky [2][iD] and Natan Kopeika [2,*]**

[1]  Department of Electrical and Electronic Engineering, Ariel University, Ariel 40700, Israel; rozbandaniel@gmail.com (D.R.); mosheg131@gmail.com (M.G.)
[2]  Department of Electro-Optical and Photonics Engineering, School of Electrical and Computer Engineering, Ben-Gurion University of the Negev, Beer Sheva 8410501, Israel; ytshak@bgu.ac.il
[\*]  Correspondence: lidor8531@gmail.com (L.K.); amir007@ariel.ac.il (A.A.); kopeika@bgu.ac.il (N.K.)

**Abstract:** A full proof of concept for low-cost millimeter wave (MMW) communication link is demonstrated in this study. The suggested MMW channel is based on a very inexpensive commercially available off-axis dish antenna usually used for TV satellites, two Arduino Uno micro controller boards, and glow discharge detectors (GDD). The GDD is a robust and inexpensive room-temperature plasma device which was found to be a sensitive MMW radiation detector. The Arduino micro controllers are used to encode a text message into serial bits and also decode it. Those serial bits were used to modulate the MMW radiation in On-Off keying. The detection of MMW radiation was performed using a simple and inexpensive GDD. The suggested MMW channel can be used as point to point backhaul wireless communication for the 5th generation of cellular communication.

**Keywords:** plasma detectors; Quasi-optical design; millimeter wave (MMW) communication

## 1. Introduction

In order to satisfy the demand for huge data rates, the carrier frequency must be increased to the X band, K band, and/or millimeter-wave (MMW) (30–300 GHz) portion of the electromagnetic spectrum [1,2]. The millimeter-wave (MMW) and terahertz (THz) bands (30 GHz to 300 GHz) are good solutions for providing high data communication rates [2,3]. Thus, the use of millimeter wave (MMW) and THz radiation for wireless communication has increased significantly. More and more applications in wireless communication, autonomic vehicles, and space technology are based on MMW and THz radiation bands [3,4]. Furthermore, inexpensive quasi optical components such as reflectors and mirrors, are necessary for the implementation of such MMW and THz communication channels [5,6]. In order to fulfill requirements for massive data traffic to core cellular network in the 5th generation of wireless communication, a fast backhaul network is required [7]. Optic fiber backhaul cannot be used in some regions due to infrastructure issues and the current microwave solution cannot fulfill the growing demands for high speed traffic channels [8].

Miniature neon indicator lamps, also known as glow discharge detectors (GDDs), were found to be very sensitive and inexpensive detectors (0.3$ each) for MMW and THz radiation [9–22]. The GDD is a room temperature detector, capable of direct and heterodyne detection [14,15,19]. In previous studies, The NEP and responsivity of the GDD were calculated for the electronic detection system at 100 GHz and resulted in 10 nW/√Hz and 31 V/W respectively [9]. MMW Focal plane arrays (FPA) based on GDD pixels were constructed and experimentally demonstrated [13]. In those demonstrations, the detection mechanism of the GDD was based on slight change of the GDD electrical current as a

function of the incident MMW/THz radiation [16–18]. Furthermore, upconversion of MMW power into visual light range using the GDD was demonstrated as well [9]. Operation of GDDs in upconversion mode for MMW coherent detection was also experimentally demonstrated in [23]. Many GDD neon lamps from different vendors were tested and the Gilway N523 was found to be the most responsive of them for electrical detection.

In this study we present a very inexpensive MMW wireless point to point (PTP) communication channel concept (see Figure 1). The quasi-optical set up of the MMW channel was composed of two inexpensive and commercially available satellite TV dish antennae. Those satellite TV dishes were coated with metal stickers in order to ensure that the metal surface is smooth enough for use in the MMW band as shown in Figure 3a–c. The Arduino micro controller board in the transmitter side is used to encode a text message into serial bits and transfer those bits to a serial port also known as a UART (TX pin). Those serial bits were used to modulate a 100 GHz carrier wave with On-Off keying (OOK) modulation. The modulated MMW beam is then coupled out to free space using a horn antenna. The horn antenna was placed in the focal plane of an off-axis dish that collimates the MMW radiation and reflects it to a second off-axis dish that focuses the radiation on the GDD cross section.

The GDD is an MMW envelope detector and it was used to detect the modulated On-OFF keying. The detected signal generated from the GDD was connected to a comparator circuit that was used to generate the proper On-OFF voltage levels that are required for detection using a second Arduino microcontroller placed on the receiver side. A computer code was used to control the Arduino microcontroller and to decode the received serial bits and present the received message. The bit rate of the transmission was set to 57,600 bps. The constructed MMW communication channel distance, $z$, was 15 m. However, larger distances can also be achieved using the same considerations. Design considerations and calculations required for realization of a very inexpensive point-to-point (PTP) MMW communication channel, in general, will be given in this study.

## 2. Quasi-optical Design of the MMW Channel, Calculations and Alignment

The whole quasi optical design for the realization of MMW communication PTP link is shown in Figure 1. This unique design is composed of two standard TV satellite dishes, 100 GHz MMW source, 2 Arduino Uno microcontroler, GDD detector circuit shown in Figure 2 and an oscilloscope. The 100 GHz source (TX272 VDI) used here was manufactured by Virginia Diodes Inc (VDI). This MMW source in the W band is based on three GaAs frequencies and has a multiplication Factor of 8 to its local oscillator frequency range of 12.625–13.625 GHz [24]. The maximum output power of that source is 600 mW at 100 GHz. The encoded message bits were generated by the user and were transmitted from the Arduino microcontroller Tx pin to the TTL modulation input of the MMW source (0 to 5 V). The maximum TTL modulation rate of this MMW source is 100 MHz.

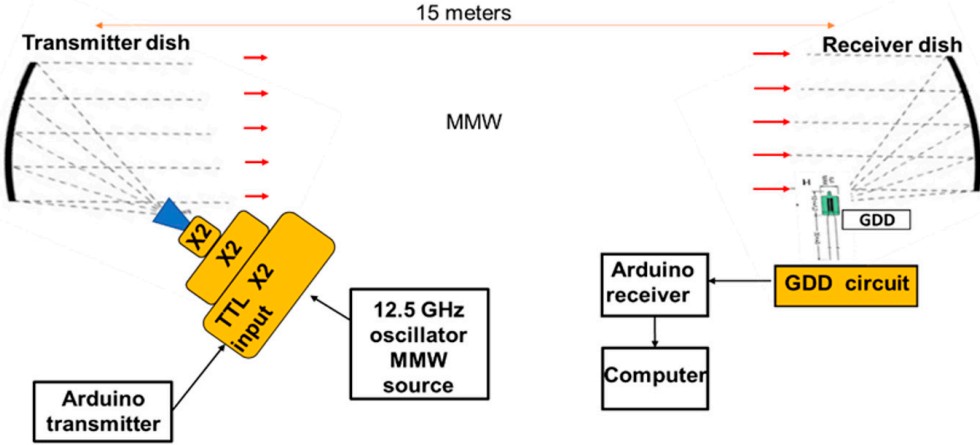

**Figure 1.** Block diagram of the experimental setup of MMW projection system.

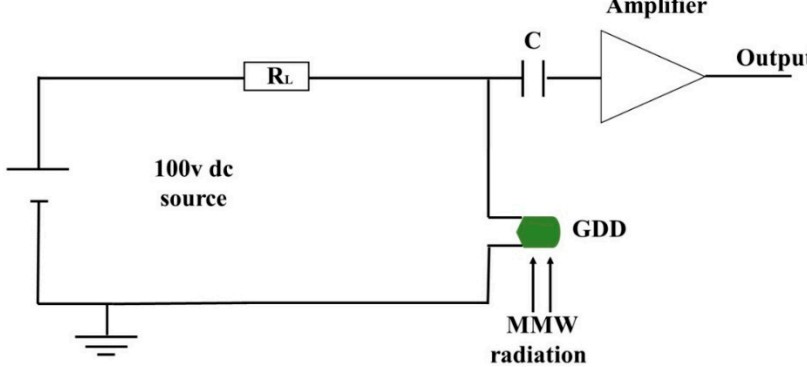

**Figure 2.** The glow discharge detectors (GDD) electrical circuit.

The GDD circuit contains an amplifier with voltage gain of 5, a dc block capacitor and a resistor $R_L$. The modulated radiation generated by the MMW source is coupled out to free space using a Virginia Diodes Conical horn antenna and a satellite dish antenna coated with aluminum. The horn antenna specifications are as follows: horn length $L_{Gh} = 35.5$ mm and aperture diameter $D_h = 16.3$ mm. The total cost of the two dish antennae and the aluminum tape is less than 50 \$. Figure 3 shows photos of those coated antennas.

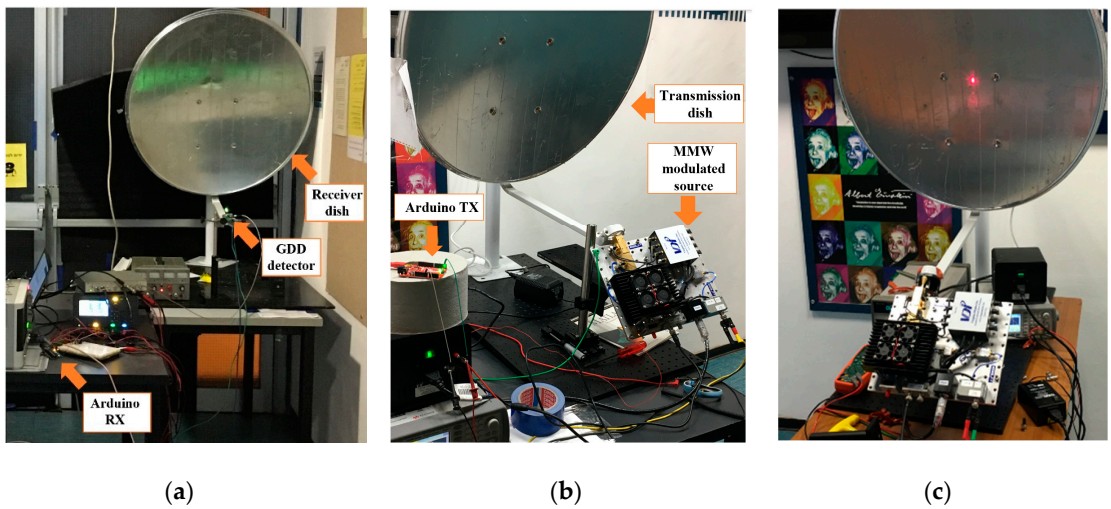

**Figure 3.** The dish antennas setup for the receiver (**a**) and transmission side (**b**) in the experiment. (**c**) Demonstration of the alignment process.

The receiver side of the MMW channel is based on a similar dish antenna. Calculations of the expected beam spot on the transmitter dish, receiver dish, and the GDD is shown next. In those calculations only the fundamental Gaussian mode (TEM$_{00}$) is being considered. This is since, according to VDI specifications for typical conical horn antenna that was used in this demonstration, 87% of total power is transmitted to free space in the fundamental Gaussian mode [25–29]. The initial alignment of the quasi-optical set up shown in Figure 3 was carried out using a laser source that was placed instead of the MMW source. The laser beam is assumed to be Gaussian with an irradiance profile that follows an ideal Gaussian distribution. Further alignments were performed by adjusting the position of quasi-optical dishes and the GDD in order to receive a stronger MMW signal. The MMW source horn antenna was located at the focal point of the transmitted dish ($F_{dish} = 600$ mm). The Gaussian

beam approximation was used to calculate the beam waist (Equation (1)), and the total power collected by the receiving dish is calculated in Equation (2).

$$\omega(z) = \omega_0 \sqrt{1 + \left(\frac{z}{z_R}\right)}, \; z_R = \frac{\pi \omega_0}{\lambda} \tag{1}$$

$$P = P_0 \left(1 - e^{-\frac{D^2}{2\omega(z)^2}}\right) \tag{2}$$

where $\omega(z)$ is beam waist, $z$ is the distance of propagation, $\lambda$ is the MMW wavelength, $Z_R$ is the Rayleigh range of a Gaussian beam, $P_0$ is the total beam power emitted from the MMW source, and $D$ is the diameter of the receiving element. The diameter of the transmitting and receiving dishes is 800 mm. Equation (2) can be used to calculate the power collected by the transmitting and receiving dishes and also by the detector (assuming a circular detecting aperture). To use Equation (2), the MMW beam waist has to be calculated in those positions.

According to VDI's nominal horn specifications the MMW beam waist radius refers to the input of the conical horn antenna and is fixed by that geometry to $\omega_0 = 6.2$ mm [25–29]. The MMW source was placed in the focal point of the transmitter dish and the GDD was placed at the focal point of the receiving dish (see Figure 1). The focal lengths of both dishes are also fixed and can't be altered and are equal to $F_{dish} = 600$ mm. The approximate beam waist can be calculated for a given position in the quasi optical setup shown in Figure 1, using the transformation of the complex beam parameter $q_1$ at the output of the MMW source ($W = 6.2$ mm and $R = \infty$) by the ABCD transfer matrix for paraxial rays. The complex beam parameter $q_2$ can then be calculated at that given position using Equation (4). Knowing the complex beam parameter $q_2$ can allow the calculation of the beam waist W and the beam radius of curvature $R$ at that given position using Equation (5). The parameters in Equation (3) are as follows:

$$\begin{bmatrix} A & B \\ C & D \end{bmatrix} = \begin{bmatrix} 1 & d_3 \\ 0 & 1 \end{bmatrix} \begin{bmatrix} 1 & 0 \\ -1/F_{dish} & 1 \end{bmatrix} \begin{bmatrix} 1 & d_2 \\ 0 & 1 \end{bmatrix} \begin{bmatrix} 1 & 0 \\ -1/F_{dish} & 1 \end{bmatrix} \begin{bmatrix} 1 & d_1 \\ 0 & 1 \end{bmatrix} \tag{3}$$

$$q_2 = \frac{Aq_1 + B}{Cq_1 + D} \tag{4}$$

$$\frac{1}{q_2} = \frac{1}{R} - \frac{i\lambda}{\pi W^2} \tag{5}$$

$d_1$ is the distance between the transmitter and the projecting dish, $F_{dish}$ is the focal length of both of the dishes, $d_2$ is the distance between the projecting and receiving dishes and $d_3$ is the distance between the receiving dish and the detector plane which is identical to the focal length of the receiving dish and equal to (600 mm). The MMW losses that are shown in Table 1 were calculated using Equation (2) with the aperture diameter D of 800 mm for the transmitting and receiving dishes and a diameter of 1mm for the effective detector aperture. To generate the final ABCD matrix shown in Equation (3), the individual matrices representing each section of the system shown in Figure 1 have to be multiplied together. The beam waist at a given point can be calculated by using the individual matrices representing each section up to that point to produce the total ABCD matrix, and then Equations (4) and (5) can be used to calculate the beam waist at that point.

**Table 1.** Approximated beam diameter and approximated percentage of total power collected by the receiving dishes are given for transmitter to receiver distances of $z = 15$ m, $z = 20$ m, $z = 50$ m and $z = 500$ m.

| Distance between transmitter and receiver [m] | 15 | 20 | 50 | 100 | 500 |
|---|---|---|---|---|---|
| Beam waist on the Transmitting dish [mm] | 92 | 92 | 92 | 92 | 92 |
| Beam waist on the receiving dish [mm] | 175 | 220 | 518 | 1031 | 5161 |
| Percentage of total power collected by the receiving dish. | 100% | 99.86% | 69% | 26% | 1.2% |
| Percentage of total power collected by the GDD detector (effective aperture diameter of 1 mm). | 1.64% | 1.62% | 1.13% | 0.42% | 0.019% |

From Table 1 it can be noticed that for up to 50 m, the suggested concept can produce a strong signal. For larger distance communication the MMW source needs to be relocated slightly further from the focal point of the dish antenna, in order to obtain greater spot size at the transmitting dish and consequently a smaller spot size at the receiving dish. In the set-up of Figure 1, the emitted MMW radiation was collimated using a dish antenna and directed to the second dish antenna that focused the beam on the effective detection area of the GDD. The current MMW technology can produce MMW power of more than 600 mW and also detect signals lower than 1 µW. Under such conditions it will be theoretically possible to transmit MMW power with the quasi-optical shown in Figure 1 for distances longer than 10 km. The dishes used here have a diameter of 800 mm. Better results can be achieved with dishes that have larger diameters, which increase the power collected by the receiving dish.

## 3. Results

Figure 4 top shows the Arduino Uno microcontroller transmitted bits. Figure 4 bottom show the GDD detected bits. A comparator circuit is used as a decision circuit that decides whether an incoming binary signal is at the logical 0 level or at the logical 1. The output voltage of this comparator is shown in the middle of Figure 4. The VDI MMW source used for the transmission of serial bits from the Arduino microcontroller had an MMW frequency of 100 GHz. The incoming binary signal was encoded using the Arduino Uno microcontroller and the message was shown using the Arduino software (serial monitor option). The SNR of the proposed system for distance between transmitter and receiver dishes of up to 20 m, depends mainly on the detector noise. The NEP and responsivity of the GDD were calculated for the electronic detection system at 100 GHz and resulted in 10 nW/$\sqrt{\text{Hz}}$ and 31 V/W respectively [9]. The MMW source transmission power was about 600 mW, and the measured beam waist was about 10 mm. Therefore the total power collected by the GDD effective cross section is about 3.8 mW and, because of the On-Off modulation, the average power is approximately 1.9 mW. The GDD circuit of Figure 2 was connected to an amplifier with gain of 5, therefore the response of the GDD can be calculated by multiplying the received power of 1.5 mW by the responsivity of 31 V/W and by the amplifier gain of 5. The calculated response is therefore approximately 295 mV. The noise voltage is approximately 310 nv/$\sqrt{\text{Hz}}$ and the bandwidth was about 1MHz, therefore the noise voltage can be calculated to be about 0.31 mV RMS or 6.2 mV RMS after amplification. The signal shown in Figure 4 is about 270 mV peak to peak voltage as was expected (95 mV RMS). The SNR is therefore equals to about 24 dB. Thus, the performance of the communication link or (Pout/Pin) is 0.91, which means that 91 percent of the transmitted signal is received at the GDD plane. The Arduino Uno microcontroller was programmed to send the same message in loops. Based on typical chart for bit error rate of different modulation method [30] including On-Off keying shown in Figure 5 the bit error rate can be estimate at about $10^{-12}$ for a signal to noise ratio of 24 dB.

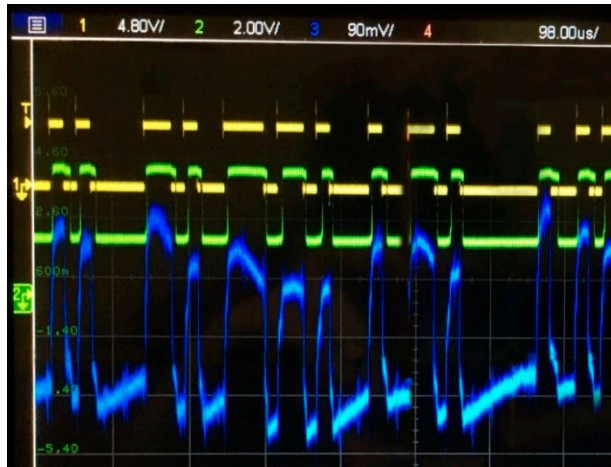

**Figure 4.** The voltage values of the Arduino transmitted bits in TTL levels (top, yellow signal), the raw detected signal received at the GDD detector of size 295 mV pick to pick' (bottom, blue signal) and the reconstructed signal of the transmitted bits in the receiver side at the comparator output in TTL levels (middle, green signal).

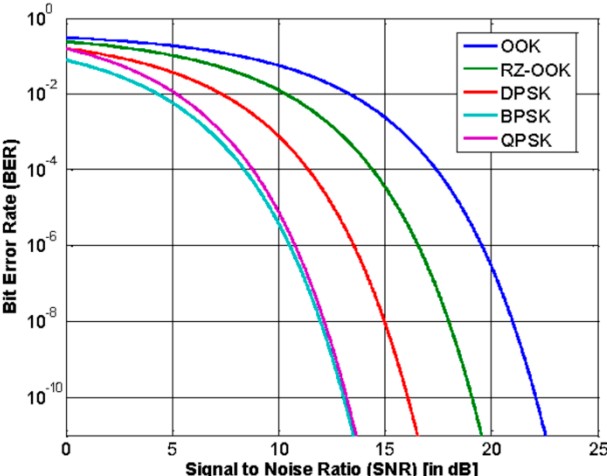

**Figure 5.** The bit error rate performance for free space communication systems under various modulation method as a function of single to noise ration.

The experimental results shown in this study prove that the proposed MMW channel can be used as a MMW point to point backhaul network. Following this proof of concept, we can now realize high data rates and a very inexpensive MMW communication channel. MMW communication channel distances of hundreds of meters can be designed and constructed based on the proposed MMW channel.

## 4. Conclusions

The experimental results of this paper show that the proposed system can serve as a very inexpensive, commercially available and easy to calibrate and operate point to point backhaul MMW wireless communication link for indoor and outdoor application. The suggested quasi-optical system can produce a strong signal with low losses depending on the distance between the transmitter and receiver, and the diameters of the dishes. The system can increase the product signal by improving the alignment of the dishes and improving the coating of the aluminum tapes to form a smoother surface. The proposed system can also be used as an inexpensive alternative for in-space satellite communications.

## 5. Patents

The GDD was patented under the title: "Upconversion system for imaging and communication", priority: US/17.07.16/ USP20166263269.

**Author Contributions:** Conceptualization, L.K., A.A. and D.R.; Methodology, L.K. and A.A.; software, D.R. and L.K.; validation, L.K., M.G. and D.R.; investigation, L.K.; resources, A.A. and N.K.; data curation, L.K. Writing—original draft preparation, L.K., D.R. and A.A.; Writing—review and editing, L.K., N.K., Y.Y.; supervision, A.A., N.K. and Y.Y.; project administration, A.A., N.K. and Y.Y.; All authors have read and agreed to the published version of the manuscript.

**Funding:** Ariel university internal funding.

**Conflicts of Interest:** The authors declare no conflict of interest.

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
