# Peer review of "Inexpensive Millimeter-Wave Communication Channel Using Glow Discharge Detector and Satellite Dish Antenna"

_electronics, doi:10.3390/electronics9040677_

Round 1
Reviewer 1 Report
The authors have presented a full proof of concept for low cost millimetre wave communication link.
Few comments:
Fig. 3 is missing in the manuscript.
Avoid using Fig 1 and Fig2. For consistency, use Fig. 3 throughout.
Authors should give more explanation on the analysis of the proposed work.
What is the impact of using different diameter size for the TX and RX dishes?
Can the authors elucidate on why the beam waist on the receiving dish were varied?
Apart from being inexpensive, what are the other novelties of the proposed work?
Author Response
The authors would like to thank the reviewers for their constructive remarks. We address and correct accordingly all the reviewer’s remarks. Attached please find our article with all remarks implemented.
reviewer 1:
1) Fig. 3 is missing in the manuscript.
indeed, this was a mistake in the figures numbering which caused the issue with Fig 3 being missing. The mistake was fixed ant the figure was added back.
2) Avoid using Fig 1 and Fig2. For consistency, use Fig. 3 throughout.
This issue was fixed throughout the article.
3)Authors should give more explanation on the analysis of the proposed work.
For a better explanation of the analysis procedure these additional equations were added as follows: Gaussian beam approximation (Eq.1), ABCD transfer matrix for paraxial rays of the quasi optical system (Eq.3), complex beam parameter q (Eq.5). More explanation on the approximate beam waist used in table 1 was added as well. See lines 112-150.
4) What is the impact of using different diameter size for the TX and RX dishes?
The impact of a bigger diameter dish was clarified. See lines 166-168.
5) Can the authors elucidate on why the beam waist on the receiving dish were varied?
The added equation 1 clarifies why once the distance Z is altered, the beam waist will also vary. See line 112.
6) Apart from being inexpensive, what are the other novelties of the proposed work?
Two additional novelties of the proposed system were added: 1) very simple to calibrate and operate 2) high performance communication link. See lines 229-231.
Reviewer 2 Report
Dear authors,
thank you for this very interesting paper. I'm glad I choose to review it. I only have some minor comments for you:
Line 68: I guess you used an Arduino Uno.
Line 73: change 100MHz to 100 MHz
Line 88: change "carried" to "carried out"
Line 92: could you please give the equation for the Gaussian beam approximation.
Line 99: please indicate if ω0 is the Gaussian beam radius (or another parameter)
Table1: please indicate how the beam waist is calculated for the two dishes
Line 115: change 10 Km to 10 km
Line 119: the Fig3 is completely missing, please add it
Line 122: place a point between "Fig.4" and "The GDD"
Line 125: change "massage" to "message" (twice on the same line)
Line 135: figure text change "massage" to "message"
That's all from me, Thank you!!
Author Response
The authors would like to thank the reviewers for their constructive remarks. We address and correct accordingly all the reviewer’s remarks. Attached please find our article with all remarks implemented.
reviewer 2:
1) I guess you used an Arduino Uno.
Line 71: Uno was misspelled and was corrected.
2) change 100MHz to 100 MHz.
The correction was made. See line 77.
3) change "carried" to "carried out".
The grammar error was corrected. See line 105.
4) could you please give the equation for the Gaussian beam approximation.
Added Equation 1 that was used for the Gaussian beam approximation. See line 112.
5) please indicate if ω0 is the Gaussian beam radius (or another parameter).
A clarification on ω0 being the radius was made. See line 123.
6) Table1: please indicate how the beam waist is calculated for the two dishes.
The calculation method for the two dishes using ABCD matrix analysis was added and clarified. See lines 123-150.
7) change 10 Km to 10 km.
The correction was made. See line 166.
8) the Fig3 is completely missing, please add it.
indeed, this was a mistake in the figures numbering which caused the issue with Fig 3 being missing. The mistake was fixed ant the figure was added back
.9) place a point between "Fig.4" and "The GDD"
done
10) change "massage" to "message" (twice on the same line), figure text change "massage" to "message".
This grammar error was fixed throughout the article.
Reviewer 3 Report
The paper describes some experimental results of a millimeter wave communication system using a GDD as a detector. Although the topic is very interesting, the work needs to be improved. Here there are some points to expand:
1)- A description, with some further details, of the millimeter wave transmitter module and the antenna used (dimensions, patterns on the principal planes)
2)- A more detailed description of the sensor used, sensitivity, noise etc.(in this case GDD, that's the core of the paper) and the block diagram and/or electrical diagram of what has been named as "GDD circuit".
3)- a link budget in order to estimate the performance of the link
4)- the quality of the images is poor, figure 4 is a screenshot of an oscilloscope in which the tracks-signals associations are not clear. To do this it is necessary to describe the operation of the "GDD circuit" block in detail.
5)- figure 5 is of low meaning, it can be explained within the paper
6)- it could be very interesting instead to try to characterize the performance of the link changing the (S/N) ratio in order to estimate the BER
Author Response
The authors would like to thank the reviewers for their constructive remarks. We address and correct accordingly all the reviewer’s remarks. Attached is the article documents with all remarks implemented.
reviewer 3:
1)- A description, with some further details, of the millimeter wave transmitter module and the antenna used (dimensions, patterns on the principal planes).
A wider description with more details of the MMW source was added, including the dimensions of the horn antenna. See lines 72-74,92-94.
2)- A more detailed description of the sensor used, sensitivity, noise etc.(in this case GDD, that's the core of the paper) and the block diagram and/or electrical diagram of what has been named as "GDD circuit".
The responsivity and noise characteristic of the GDD in electrical detection method were added. Also an additional figure of the GDD electrical circuit (Fig. 2) was added, that gives a better and more detailed description of our setup. See lines 38-39, 83-91.
3)- a link budget in order to estimate the performance of the link.
The performance of the communication link (Pout/Pin) was explained in in a clearer manner. See lines 191-192.
4)- the quality of the images is poor, figure 4 is a screenshot of an oscilloscope in which the tracks-signals associations are not clear. To do this it is necessary to describe the operation of the "GDD circuit" block in detail.
Done. See comment 2.
5)- figure 5 is of low meaning, it can be explained within the paper.
Done.
6)- it could be very interesting instead to try to characterize the performance of. the link changing the (S/N) ratio in order to estimate the BER
As specified in the article a BER performance for free space communication systems under various modulation method as a function of single to noise ration chart was added (Fig. 5) , based on the chart, the bit error rate of the on\off keying modulation method is about 10-12 for a single to noise ratio of 24dB . See lines 193-195,211-220.
Round 2
Reviewer 1 Report
The authors said "once the distance Z is altered, the beam waist will also vary". Can the authors clarified why the beam waist of the transmitting dish was fixed and not affected by Z but only the receiving dish waist?
Other comments have been fully addressed.
Author Response
The authors would like to thank the reviewers for their constructive remarks. We address and correct accordingly all the reviewer’s remarks. Attached please find our detailed corrections to the manuscript.
reviewer 1:
1)The authors said "once the distance Z is altered, the beam waist will also vary". Can the authors clarified why the beam waist of the transmitting dish was fixed and not affected by Z but only the receiving dish waist?
For better explanation of the subject we explain that both the beam waist radius of the conical horn antenna and the focal length of both dishes is fixed by the manufacturers and cannot be altered, therefore the beam waist is constant for the transmitter dish. See lines 122-131

Reviewer 3 Report
Dear Authors,
the quality of the paper has improved and suggestions have been followed. There are few little open points:
- line 194-->On-Off Keying
- improve the quality of Fig.4 (maybe using 600 DPI eps format ?) adding clear references to the traces
- improve the quality of Fig. 5
Regards,
Author Response
The authors would like to thank the reviewers for their constructive remarks. We address and correct accordingly all the reviewer’s remarks. Attached please find our detailed corrections to the manuscript.
reviewer 3:
- line 194-->On-Off Keying.
the required change has been done.
- improve the quality of Fig.4 (maybe using 600 DPI eps format ?) adding clear references to the traces.
Figure 4 quality was improved to 600 dots per inch as requested, also a more detailed review on each of the traces was added to the figure description. See lines 196-210.
- improve the quality of Fig. 5.
Figure 5 quality was improved. See lines 212-218.
